

# Relationships between neuropsychological and antisaccade measures in multiple sclerosis patients

Marisa Borges Ferreira[1,2], Paulo Alexandre Pereira[2,3], Marta Parreira[2], Ines Sousa[4], José Figueiredo[2], João José Cerqueira[5,6] and Antonio Filipe Macedo[1,7]

[1] Low Vision and Visual Rehabilitation Lab, Department and Center of Physics—Optometry and Vision Science, University of Minho, Braga, Portugal
[2] Association "Todos com a Esclerose Multipla (TEM)", Braga, Portugal
[3] Centre of Mathematics and Department of Mathematics and Applications, University of Minho, Braga, Portugal
[4] Centre of Molecular and Environmental Biology (CBMA), and Department of Mathematics and Applications, Universidade do Minho, Braga, Portugal
[5] Neurosciences Domain; Life and Health Sciences Research Institute, School of Health Sciences and ICVS/3B's Associate Laboratory, University of Minho, Braga, Portugal
[6] Clinical Academic Centre (CCA), Hospital de Braga, Braga, Portugal
[7] Department of Medicine and Optometry, Linnaeus University, Kalmar, Sweden

Corresponding author
Antonio Filipe Macedo,
antonio.macedo@lnu.se

## ABSTRACT

**Background:** The Stroop test is frequently used to assess deficits in inhibitory control in people with multiple sclerosis (MS). This test has limitations and antisaccade eye movements, that also measure inhibitory control, may be an alternative to Stroop.
**Objectives:** The aim of this study was twofold: (i) to investigate if the performance in the antisaccade task is altered in patients with MS and (ii) to investigate the correlation between performances in neuropsychological tests, the Stroop test and the antisaccade task.
**Methods:** We measured antisaccades (AS) parameters with an infrared eye tracker (SMIRED 250 Hz) using a standard AS paradigm. A total of 38 subjects diagnosed with MS and 38 age and gender matched controls participated in this study. Neuropsychological measures were obtained from the MS group.
**Results:** Patients with MS have higher error rates and prolonged latency than controls in the antisaccade task. There was a consistent association between the Stroop performance and AS latency. Stroop performance but not AS latency was associated with other neuropsychological measures in which the MS group showed deficits.
**Conclusions:** Our findings suggest that AS may be a selective and independent measure to investigate inhibitory control in patients with MS. More studies are necessary to confirm our results and to describe brain correlates associated with impaired performance in the antisaccade task in people diagnosed with MS.

## INTRODUCTION

Multiple sclerosis (MS) is a neurodegenerative disease producing inflammation of the central nervous system characterised by loss of oligodendrocytes and axonal degeneration (*Keegan & Noseworthy, 2002*). Cognitive deficit is often the first indicator of the disease progression and can involve a variety of functions such as processing speed, long-term memory, attention and executive functions (*Chiaravalloti & DeLuca, 2008*; *Patti et al., 2009*; *Ruggieri et al., 2003*). Executive functions involve, amongst others cognitive processes, inhibitory control. Inhibitory control can be defined as the ability to suppress prepotent but incorrect responses and the ability to filter out irrelevant information within a stimulus set (*Botvinick et al., 2001*). Prepotent response inhibition summons processes that regulate the selection of a weaker, but task-relevant response, over a competing strong, but task-irrelevant, response (*Miller & Cohen, 2001*). Antisaccades (AS) performance obtained with an AS task is considered a good indicator of inhibitory control in humans. For example, performance on this task is correlated with neuropsychological tests of executive functions in multiple neurodegenerative diseases (*Heuer et al., 2013*; *Hutton, 2008*).

The AS task requires suppression of an automatic saccade towards a sudden-onset peripheral visual target (defined as prosaccade) and the generation of a volitional (endogenous) saccade towards the mirrored target location (antisaccade) (*Hutton & Ettinger, 2006*; *Munoz & Everling, 2004*; *Olk & Kingstone, 2003*; *Talanow et al., 2016*). In humans the integrity of the dorsolateral prefrontal cortex (DLPFC) and anterior cingulate gyrus (ACC) seems to be critical for good AS performance (*Pierrot-Deseilligny et al., 2003*). Animal studies have shown that neural activity in correct AS (looking away from the target) in regions such as the prefrontal and anterior cingulate cortices is higher than when a wrong decision (prosaccade, or looking at the target) is made (*Amador, Schlag-Rey & Schlag, 2004*; *Johnston & Everling, 2011*).

Programming AS requires inhibitory and generation phases involving activation of the ventrolateral and dorsolateral cortices (*Asaad, Rainer & Miller, 2000*; *Miller & Cohen, 2001*). These areas are also involved in, for example, decision making (*Ettinger et al., 2008*). A detailed description of the process can be found in the referred literature. In brief, the structures involved are thought to be mediating processes of vector transformation—for AS the locations of the stimulus and the saccadic goal are not the same and, in the transition from stimulus encoding to the initiation of the response, the stimulus vector must be inverted 180° into the movement vector (*Moon et al., 2007*; *Talanow et al., 2016*).

There are different opinions concerning the cognitive mechanisms underlying AS performance, a good summary can be found in a publication by *Talanow et al. (2016)*. In his publication Talanow describes a theoretical model that is currently accepted to explain AS performance in terms of parallel programming, often linked to a competition model or the horse-race metaphor (*Hutton, 2008*; *Hutton & Ettinger, 2006*; *Kristjansson, 2007*; *Munoz & Everling, 2004*; *Talanow et al., 2016*). According to this model prosaccade and AS signs are generated simultaneously in response to a visual stimulus.

Both signs compete until one of them wins the competition. Good AS performance thus requires efficient cognitive processes that are sensitive to changes in brain structure occurring in neurodegenerative diseases or with age (*Heuer et al., 2013*; *Lemos et al., 2016*; *Pa et al., 2014*).

Some studies looked at cross-sectional (*Fielding et al., 2009*) and longitudinal alterations (*Fielding et al., 2012*) in AS performance in people diagnosed with MS. These studies found that in people with MS the number of errors and latency are associated with general cognitive functioning. Others observed that AS performance was associated with structural and microstructural changes in the cerebellum (*Kolbe et al., 2014*). However, given that performance in AS tasks rely strongly in inhibitory control, we hypothesise that performance in this task may be associated with neuropsychological tests such as the Stroop test that relies in a similar type of cognitive control.

Like the AS task, the Stroop task requires overriding of the prepotent response or prepotent response inhibition. In a Stroop colour-naming task participants have to name the ink colour of a printed colour word. Greater conflict occurs for incongruent (e.g. the word **red** in **green** ink) than for congruent (e.g. the word **red** in **red** ink) trials—this is the Stroop interference effect (*Vanderhasselt, De Raedt & Baeken, 2009*). Brain correlates indicate that the cognitive control of this task involves cortico-subcortical circuit and that includes the ACC and the DLPFC (*Carter & Van Veen, 2007*; *MacDonald et al., 2000*). A response conflict is generated when there is a coactivation of incompatible stimulus-response processes, the conflicting signal is used to recruit cognitive control in order to reduce the conflict and improve performance (*Carter & Van Veen, 2007*; *Johnston & Everling, 2011*). According to the prominent conflict-monitoring hypothesis the occurrence of response conflict is signalled by the ACC leading to recruitment in the DLPFC of more cognitive control for subsequent performance (*Botvinick et al., 2001*; *Vanderhasselt, De Raedt & Baeken, 2009*).

The current neurophysiological tests such as the Stroop test may have limitations to assess inhibitory control. For example, Stroop interference performance can be related with reading ability (*Protopapas, Archonti & Skaloumbakas, 2007*). Also, some studies reported effects of intelligence and education in this test (*Homack & Riccio, 2004*) and others failed to show differences in performance between clinical groups and controls even if differences were supposed to occur (*Schwartz & Verhaeghen, 2008*). Another known factor that can interfere with the Stroop test performance are colour vision defects that occur, for example, with normal aging or as a result of diseases such as MS (*Ben-David & Schneider, 2010*; *Martinez-Lapiscina et al., 2014*). These facts inspired us to investigate if the AS task can be a reliable test that overcomes some of the known limitations of the Stroop test but still provides a fine assessment of inhibitory control in people with MS (*Heuer et al., 2013*; *Sisco et al., 2016*).

The aim of this study was to investigate AS performance in patients with MS and the association between AS and Stroop performances. We formulate two hypotheses: (i) the AS task is sensitive to capture subclinical functional impairment in inhibitory control in people with MS and (ii) AS performance is correlated with the performance in the Stroop task.

## METHODS

### Participants

A total of 38 patients diagnosed with MS were recruited by one of the authors (João José Cerqueira) at Hospital de Braga. An equal number of controls, age and gender matched, was recruited. The inclusion criteria for the MS group were: age between 18 and 45 years old, relapsing remitting course, early stages of disease severity as measured by the Expanded Disability Status Scale (EDSS) ≤3 (*Kurtzke, 1983*), and normal or corrected to normal visual acuity. Exclusion criteria were: on-going relapse/relapse in the previous month, presence of clinically diagnosed cognitive impairment, history of traumatic brain injury and/or stroke, depression (self-reported or detected during the study using the Beck Depression Inventory (BDI) (*Beck, Steer & Carbin, 1988*)). Those with clinically visible oculomotor abnormalities were not recruited (e.g. nystagmus or internuclear ophthalmoplegia, see references (*Frohman et al., 2005*; *Serra, Chisari & Matta, 2018* (Table 1); *Serra et al., 2003*) for comprehensive lists. Controls were subject to equivalent exclusion criteria. The ethics committee at University of Minho granted ethical approval to carry out the study, ref: SECVS 083-2013, and all participants were informed about the aim and procedures involved before signing the informed consent. For a summary of demographic characteristics see Table 1 and for neuropsychological results see Table 2.

### Equipment and procedures

Eye movements were monitored using a binocular eyetracker running at 250 Hz (RED250; SMIGmb, Teltow, Germany). The eyetracker has a spatial resolution <0.4°, is controlled with iView X software (v2.8, August 2011; SMIGmb, Teltow, Germany) and stimuli were presented on a 22 inch (1,680 × 1,050 px) LCD monitor running at 60 Hz (Dell P2210). The system used was formed of two computers connected by a high-speed Ethernet. One computer controls the eyetracker and other controls the stimulus presentation. A Matlab software development kit provided by SMI and elements of the Psychophysics toolbox were used for running the experiment (*Pelli, 1997*). Code for data analysis was also written in Matlab v2010b (The MathWorks, Inc., Natick, MA, USA).

Experiments were performed in a dimly lit room (~10 lux) and stimulus Michelson contrast was 90%. One block of eight practice trials were performed before data collection to make participants familiar with calibration and with the task, a second block of practice was given when participants did predominantly errors in the practice trials or expressed difficulties to understand their task. A five-point calibration procedure was applied, only participants with calibration accuracy (mean deviation for the expected position) of 1° or less in both $x$-axis and $y$-axis were considered. Participants were seated 70 cm from the monitor with their head restrained with a headband attached to the seat.

The paradigm shown in Fig. 1 consisted of two steps. Step 1: subjects had to fixate a centrally located target (cross) during a variable interval of 1,250 or 1,600 ms, intervals were assigned in a random order. The fixation target for step 1 was only visible after fixating a gaze-contingent box for 150 ms. Step 2: after the interval defined for step 1,

**Table 1 Summary of the demographic and clinical information of the final sample of participants for both groups (SD = standard deviation).**

| Variable | Group | |
| --- | --- | --- |
| | **MS (n = 38)** | **Control (n = 38)** |
| Gender (n (%)) | 24 females (63%) | 21 females (55%) |
| Age (mean (SD)) | 37 (6) | 36 (6) |
| Time since diagnosis in months (mean (SD)) | 96 (59) | – |
| EDSS (median (IQR)) | 1.5 (2) | – |

Note:
Differences in age and gender were not statistically significant.

**Table 2 Summary of the psychological evaluation results of the final sample of MS patients and normalised data for the Portuguese population (SD = standard deviation).**

| Variable | Group | | |
| --- | --- | --- | --- |
| | **MS (n = 38)** | **Normative** | ***p*-value** |
| | **Mean (SD)** | **Mean (SD)** | |
| MOCA total | 24.03 (2.7) | 26.4 (2.2) | *p < 0.001* |
| Digit span (WAIS-III) | 9.8 (2.2) | 10 (3) | 0.68 |
| Trail making (women) | 111 (64) | 61 (37) | *p = 0.001* |
| Trail making (men) | 108 (54) | 52 (37) | *p = 0.002* |
| SDMT (women) | 51.61 (8.4) | 60 (10.1) | *p < 0.001* |
| SDMT (men) | 51.07 (10.7) | 65.9 (12.2) | *p < 0.001* |
| COWAT | 32.3 (10.25) | 33.2 (9.6) | 0.60 |
| Twenty questions (D-KEFS) | 10.5 (3.3) | 10 (3) | 0.35 |

a peripheral target was presented at 5° or 10° of visual angle at right or left side of the fixation target, all locations were assigned in random order. Before data collection and during practice trials subjects were clearly instructed to look to the empty side of the screen to a position equivalent to where the target was visible: an AS. They were also instructed to perform this movement as quickly as possible. Each subject performed 40 trials and the duration of the task was approximately 3 min.

## Neuropsychological tests

A battery of tests was chosen to characterise the cognitive status of our participants with MS. The neuropsychological tests used were: (1) BDI (*Beck, Steer & Carbin, 1988*; *Serra & Abreu, 1973*); (2) Montreal Cognitive Assessment (*Freitas et al., 2010*; *Nasreddine et al., 2005*); (3) Digit Span (WAIS-III: Wechsler Adult Intelligence Scale—Version III) and (4) Trail Making Test, Part A (*Bucks, 2013*; *Cavaco et al., 2013*; *Seabra-Santos et al., 2003*; *Tombaugh, 2004*; *Wechsler, 1997*); (5) Symbol Digit Modalities Test (SDMT) (*Da Costa Pinto, 2004*; *Smith, 1968*); (6) Controlled Oral Word Association Test (COWAT) for phonetic fluency (*Cavaco et al., 2013*; *Heaton et al., 2004*; *Tombaugh, Kozak & Rees, 1999*); (7) A total of 20 Questions Test from Delis-Kaplan Executive Function System (DKEFS 20Q) (*Delis, Kaplan & Kramer, 2001*; *Stephens, 2014*); (8) Stroop Colour-Word test. All neuropsychological tests have normative
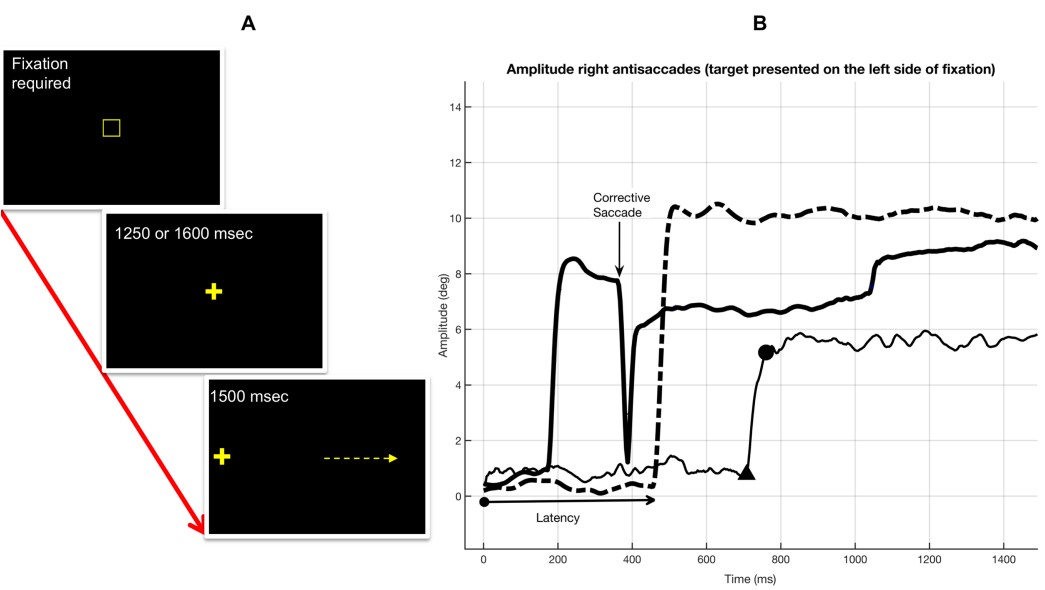

**Figure 1 The antisaccade paradigm and position traces of the eyes.** (A) The antisaccade paradigm. Step 1: subjects had to fixate a centrally located target (cross, 30 × 30 mm, ref 23) during an interval of 1,250 or 1,600 ms which assigned in a random order. The fixation target for step 1 was only visible after fixating a gaze-contingent box (10 × 10 mm) for 150 ms. Step 2: after the interval defined for step 1, a peripheral target was presented at 5° or 10° of visual angle at right or left side of the fixation target, all locations were assigned in random order (the arrow points the direction of the expected antisaccade). (B) Representation of changes in eye movement amplitude with time during three trials. Antisaccades where performed to the right when the target was presented at 10° from the central fixation target on left side of the screen. The thick-solid line shows a trial in which a corrective saccade was necessary after an initial error—amplitude has been computed using *xy* coordinates and that is why it never reaches 0° during the corrective saccade. The dotted line shows a correct AS with a good spatial precision when compared with the mirrored position of the target. The thin-solid line shows a correct AS with a large latency and poor spatial precision. In the thin-solid line the triangle shows the instant at which the AS was detected and the dot shows the instant when the AS ended.      

data for the Portuguese population (tests have been translated into the Portuguese language and administered to native Portuguese speakers), results from the MS group are compared with the normative values in Table 2. These tests were not performed by controls and were performed by MS participants after the AS task.

The final score of the Stroop colour-word test corresponds to the number of correctly reported colours in 45 s, the maximum possible score is 100 (*Castro, Martins & Cunha, 2003*; *Golden, 1975*). We scored the Stroop test using different methods that have been described by others (*Denney & Lynch, 2009*; *Sisco et al., 2016*) and formulas are specified in Table 3. We used different formulas to cover different scenarios because there are reports in the literature showing that two different formulas applied to the same data can lead to opposite conclusions (*Denney & Lynch, 2009*; *Sisco et al., 2016*).

## Data analysis

Eye movements were collected binocularly with the point-of-regard (POR) information being: $(POR_{right} + POR_{left})/2$, that is the mean POR of both eyes. AS latency is shown in Fig. 1B and represents the period of time from target onset until the reaction of the eyes

**Table 3 Summary of Stroop results for the multiple sclerosis group.**

| Stroop test score | Values | |
|---|---|---|
| | **Mean (SD)** | **Median (IQR)** |
| Word reading (W) | 90.97 (13.01) | 95.00 (5.00) |
| Colour naming (C) | 69.50 (9.99) | 69.00 (5.75) |
| Interference | | |
| Raw (CW = correct colours in 45 s) | 35.50 (9.04) | 38.00 (5.75) |
| Golden = $((W \times C)/(W + C)) - CW$ | −3.35 (7.50) | −0.55 (3.85) |
| Relative = $((C - CW/C) \times 100$ | 48.08 (10.47) | 43.28 (7.16) |
| Ratio = $(C/CW) \times 10$ | 19.54 (3.74) | 17.63 (2.56) |

**Note:**
   Formulas for each score are specified in the table (SD = standard deviation, IQR = interquartile range).

moving towards the mirrored target location. For analysed trials, AS detection was performed after smoothing $xy$ positions with a five-sample moving average. This method has been proposed and used to reduce noise (*Engbert & Kliegl, 2003*; *Ferreira et al., 2017*). After smoothing, eye velocity and acceleration were computed, saccades were detected using the velocity threshold of 30°/s and/or acceleration threshold of 8,500°/s$^2$ with a minimum duration of 12 ms (*Macedo, Crossland & Rubin, 2008*, *2011*), microsaccades were ignored (*Martinez-Conde, Macknik & Hubel, 2004*).

In addition to the number of errors and peak-velocity, three latencies were computed: (1) AS, (2) prosaccades and (3) corrective saccades. When the first saccade is made towards the target the movement is defined as a prosaccade and corresponds to an error. When after a prosaccade participants made a saccade away from the target this is defined as corrective saccade, the interval between the prosaccade and the corrective saccade is defined as corrective saccade latency. Corrective saccades were considered when its amplitude was 2° or more and performed away from the target.

## Statistical analysis

The normality of the variables was tested using the Kolmogorov–Smirnov test. Differences between neuropsychological results and normative values for the Portuguese population were verified with $t$-test, the difference in number of errors between groups was evaluated with $t$-test as well.

The effect of group, direction and eccentricity on AS parameters was tested with linear mixed models (LMM) in SPSS (IBM SPSS Statistics for Windows, Version 22.0. IBM Corp, Armonk, NY, USA) and R (lme4 package) (*Bates et al., 2015*; *Ferreira et al., 2017*; *Kliegl et al., 2010*). For this analysis eye movements parameters reported were defined as 'dependent variable' or 'response variable of interest' (e.g. latency). Participants were defined as 'random factors' or 'group specific effects.' Explanatory factors or 'fixed factors' were: 'group' (MS and Control), 'direction' (right and left), 'eccentricity' (5° and 10°) and 'saccade type' (antisaccade, prosaccade and corrective). Bonferroni correction was applied for multiple comparisons. Means described in the text and shown in graphs are the

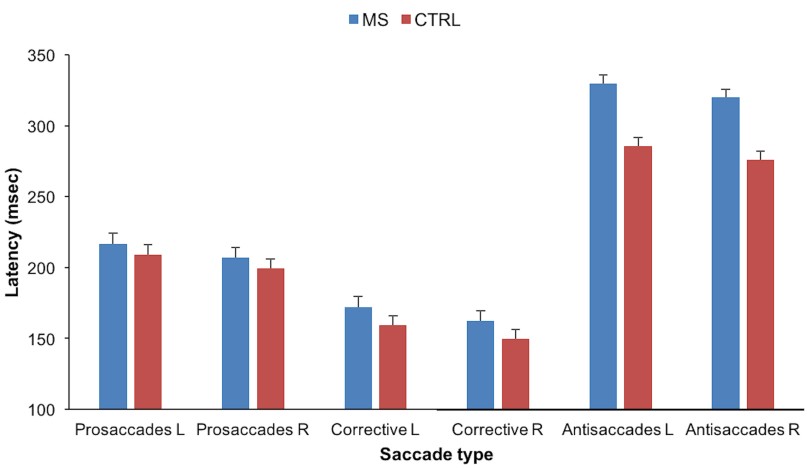

**Figure 2 Bar graph showing the latency values in milliseconds per saccade type for both groups and for both directions of the expected antisaccades.** Directions indicate the direction of the expected antisaccades (R directions the target was presented left and vice versa for L directions). Bars represent the EMM (mean response for each factor, adjusted for any other variables in the model) and the error bars the SE of the EMM for the specified factors.

estimated marginal means (EMM = mean response for each variable, adjusted for any other variables in the model) and their standard errors (SEs) for the specified factors. Correlations between neuropsychological tests and AS parameters were also tested.

## RESULTS

Demographic characteristics of the 38 participants for each group are summarised in Table 1. Table 2 summarises the comparisons between the normative values for the neuropsychological testes and the results for our participants. Table 3 summarises the Stroop test results scored in accordance with different formulas.

### AS errors

There was no effect of eccentricity in any of the AS results analysed and therefore results for 5° and 10° are reported together. The mean proportion of directional errors was 26% (SD = 18) for the MS group and 16% (SD = 11) for the control group. The mean difference of 10% in the number of errors between the MS group and the control group was statistically significant, $t(60) = 2.8$ ($p = 0.007$, unequal variances considered).

### AS latency

Latency results are shown in Fig. 2. A model (LMM) with four fixed factors was run (group: patients or controls; saccade type: AS, prosaccades or corrective; eccentricity: 5° or 10°; direction of the eye movement: left or right) to compare latencies. Only statistically significant effects are reported. There was a main effect of group $F(1, 88) = 6.4$ ($p = 0.013$) and an interaction group × saccade type $F(2, 2382) = 24$ ($p < 0.001$). There was also a main effect of saccade type $F(2, 2382) = 1,298$ ($p < 0.001$) and trial direction $F = (1, 2367) = 25$ ($p < 0.001$). The directional effect has been reported before and is likely to be explained by the higher frequency of right-eye dominance (*Vergilino-Perez et al., 2012*).

**Table 4 Correlations between the Stroop test and other neuropsychological measures ('Test' column).**

| Test | Measure | Stroop | | Stroop interference | | | |
|---|---|---|---|---|---|---|---|
| | | Word reading | Colour naming | Raw | Golden | Relative | Ratio |
| MOCA ($n = 38$) | General | $r = 0.32$ $p = 0.05$ | ns | $r = 0.42$ $p = 0.01$ | ns | ns | $r = -0.32$ $p = 0.050$ |
| WAISIII digit span ($n = 28^{\Phi}$) | Memory attention | ns | ns | $r = 0.44$ $p = 0.018$ | ns | ns | ns |
| Trail making A ($n = 37$) | Attention | $r = -0.33$ $p = 0.05$ | ns | ns | ns | ns | ns |
| SDMT ($n = 37^{\Phi}$) | Processing | ns | ns | $r = 0.48$ $p = 0.003$ | $r = 0.37$ $p = 0.024$ | $r = 0.36$ $p = 0.028$ | ns |
| COWAT ($n = 38$) | Executive | ns | ns | $r = 0.37$ $p = 0.021$ | $r = 0.32$ $p = 0.052$ | $r = -0.47$ $p = 0.003$ | $r = -0.33$ $p = 0.042$ |
| DKEFS 20Q ($n = 37^{\Phi}$) | Executive | ns | ns | ns | ns | ns | ns |

**Note:**
[Φ] When $n$ is less than 38 is because there was missing data from the neuropsychological test. Executive = Executive function; General = General cognitive function; Processing = Processing speed measure. "ns" = the correlation is not statistically significant.

Antisaccades latency was higher in the MS group than in controls, the EMM in the MS group was 325 ms (SE = 6.0) and 280 ms (SE = 6.0) in the control group. The difference between means was 45 ms, $t(2,362) = 5.1$ ($p < 0.001$). The latency of prosaccades was different from the latency of corrective saccades and both were different from the latency of AS. The mean latency for prosaccades was 209 ms (SE = 5.0), for corrective saccades was 161 ms (SE = 5.0), the difference between means was 47 ms, $t(2,362) = -8.4$ ($p < 0.001$). The mean difference between eye movements performed to the right (target presented at left) and to the left (target presented at right) was 10 ms, $t(2,362) = -5.3$ ($p < 0.001$). Differences in latency for prosaccades and corrective saccades between groups were not statistically significant, more details are given in supplemental files.

### AS peak-velocity

A model similar to that reported for latency was used to determine differences in peak-velocity. Here we found two main effects: saccade type $F(2, 2377) = 98$ ($p < 0.001$) and trial direction $F(1, 2366) = 24$ ($p < 0.001$). The mean difference in peak-velocity between right (EMM = 288°/s, SE = 7.0) and left (EMM = 275°/s, SE = 7.0) AS was 13°/s, $t(2,362) = -4.84$ ($p < 0.001$). Corrective saccades (EMM = 321°/s, SE = 7.8) showed higher peak-velocities than prosaccades (EMM = 260°/s, SE = 7.9) and AS (EMM = 263°/s, SE = 6.9). The mean difference between prosaccades and corrective saccades was 60°/s, $t(2,362) = -11.51$ ($p < 0.001$) and between corrective and AS was 58°/s, $t(2,362) = 13.81$ ($p < 0.001$). Differences in peak-velocity, for all types of saccades, between groups were not statistically significant. More details are given in supplemental files.

### Correlations between neuropsychological tests and antisaccades

Correlations between Stroop test scores and neuropsychological tests scores are summarised in Table 4. The neuropsychological tests are categorised for the main cognitive function assessed.

**Table 5 Correlation between antisaccades measures and Stroop interference scores.**

| Anti-saccades | Stroop interference | | | |
|---|---|---|---|---|
| | Raw | Golden | Relative | Ratio |
| Errors | ns | ns | ns | ns |
| AS latency | $r = -0.35$, $p = 0.034$ ($n = 38$) | $r = -0.33$, $p = 0.044$ ($n = 38$) | $r = -0.41$, $p = 0.01$ ($n = 38$) | $r = -0.38$, $p = 0.018$ ($n = 38$) |
| Prosaccade latency | ns | ns | $r = -0.33$, $p = 0.039$ ($n = 38$) | ns |
| Corrective latency | $r = -0.43$, $p = 0.011$ ($n = 35$) | ns | ns | ns |

Note:
Errors, prosaccades and corrective saccades are defined in the methods section.

Correlations between Stroop test and latency of AS, prosaccades and corrective saccades, are summarised in Table 5. The correlations between scores of all other neuropsychological tests, AS errors and latency were not statistically significant.

# DISCUSSION

Deficits in inhibitory control, that is, the ability to suppress prepotent but incorrect responses, are expected in people diagnosed with MS. The Stroop interference task and the antisaccade task both require good inhibitory control and; therefore, the performance of patients with MS in these tasks is likely to be impaired (*Carter & Van Veen, 2007*; *Johnston & Everling, 2011*). In this study we hypothesise that the antisaccade task can be a quantitative surrogate marker of neural damage leading to impairment in inhibitory control in people with MS. In the present study we investigated whether Stroop and AS performances are altered in people with MS and if both performances are correlated. In line with our hypotheses AS and Stroop performances were impaired and correlated. These findings are in agreement with other studies and are particularly relevant because the disease in our group of patients was at early stages—median EDSS 1.5 in a 0–10 scale, see Table 1 (*Kolbe et al., 2014*; *Ting et al., 2015*). The AS task remains a research tool because measuring and analysing eye movements requires specialised equipment and training that still note available to use in clinics. Also, the extraction of results is time-consuming and data recording protocols need standardization (*Antoniades et al., 2013*). Although, we must acknowledge that we found the results reported in the literature, from different laboratories and acquired with different equipment, remarkably consistent. For a good review of the clinical utility of AS task and other eye movements metrics in MS we recommend a recent review by Serra (*Serra, Chisari & Matta, 2018*).

Our results show a consistent association between the interference Stroop test performance and AS latency. Of note, Stroop performance—but not AS latency—was associated with other cognitive measures such as processing speed (measured with the SDMT) and verbal fluency/executive function (measure with COWAT) (*Whiteside et al., 2016*). These findings suggest that the Stroop performance can be influenced by

unspecific cognitive deficits (*Denney & Lynch, 2009*). In contrast, AS performance in our study was not correlated with cognitive, visual and socio-demographic aspects. Nevertheless, the AS task is likely to be affected by reduced visual acuity, poor comprehension of the instructions or slow cognitive response. To our knowledge, our study is the first to show the association between Stroop and AS tasks in patients with MS. Although, there is a recent study by Ting showing a equivalent association in patients with mild traumatic brain injury and concussion (*Ting et al., 2015*). Previous studies also pointed that different methods to compute Stroop scores would lead to different conclusions about the cognitive status of patients with neurological diseases (*Sisco et al., 2016*). To cover this aspect, our comparison included Stroop interference scores computed using different formulas. Notwithstanding the approach tested, the association between Stroop interference scores and AS latency was consistent, further suggesting its biological significance. Below we discuss in detail the consistency and the implications of our AS findings.

AS performance was worse in patients with MS than in controls. These results are in line with other studies investigating AS in MS (*Fielding et al., 2012, 2009*; *Kolbe et al., 2014*) and other neurological diseases (*Crawford et al., 2005*; *Peltsch et al., 2008*; *Ting et al., 2015*). We noticed that the number of directional errors in our control group was higher than the values reported by Fielding that reported only about 10% (*Fielding et al., 2009*). However, according to Hutton, healthy individuals can have a proportion of directional errors up to 20% in AS tasks (*Hutton, 2008*). In our study, the latency values in both groups are in good agreement with previous studies (*Aichert et al., 2012*; *Fielding et al., 2009*). Increased latency in the MS group can be explained considering the competition model described before. Poor performance in the AS task in MS is likely to occur because of a slow cognitive response that is due to increased neural recruitment that is necessary to compensate for reduced neural transmission (*Amador, Schlag-Rey & Schlag, 2004*; *Chiaravalloti & DeLuca, 2008*; *Crawford et al., 2002*). We speculate that MS is reducing the cognitive ability to process the vector inversion (suppression a prosaccadic reflex (*Edelman, Valenzuela & Barton, 2006*)) to program a volitional saccade. The possible consequences of a reduced cognitive ability to process the vector inversion are: (1) an increased time to perform the AS or (2) a prosaccade wins the competition (error). Impaired neural transmission that increases latency and probably also error rates is likely to occur in areas of the frontal cortex such as DLPFC and ACC (*Amador, Schlag-Rey & Schlag, 2004*; *Johnston & Everling, 2011*; *Miller & Cohen, 2001*). However, some studies mention lesions in other areas of the brain associated with reduced AS performance (*Kolbe et al., 2014*; *Ting et al., 2015*).

Differences between groups for prosaccades and corrective saccades latency and peak-velocities (for all types of saccades) were not statistically significant. Our interpretation of a lack of statistically significant differences for the latency is that—the generation of reflexive prosaccades and corrective saccades was not compromised in our MS participants. Our findings for controls, are consistent with the

results from Evdokimidis in which the mean value of prosaccades latency from a sample of 2,006 young men was 208 ms (*Evdokimidis et al., 2002*) and corrective saccades is typically 130 ms (*Tatler & Hutton, 2007*). These results are in line with results from other studies in MS (*Fielding et al., 2009*) and in other neurological diseases (*Crawford et al., 2005*). The lack of statistically significant differences in peak-velocities between groups for all types of saccades (AS, pro and corrective saccades) is reassuring because this indicates that internuclear ophthalmoplegia was unlikely in our participants with MS (*Bird & Leech, 1976*). Conversely, the difference in peak-velocities between AS and, for example, corrective saccades was obtained as expected. Corrective saccades are associated with larger amplitudes and it is known that peak-velocities increase with amplitude according with a relationship defined by the main sequence (*Leigh & Kennard, 2004*). In summary, latencies for prosaccades and corrective saccades and peak-velocities indicate that the integrity of the oculomotor plant was well preserved in our participants with MS.

One limitation of the current study is the lack of experimental data measuring neurocognitive performance in controls and that reduces the strength of the neurocognitive comparisons between groups. It has also been pointed during revision that the validity of our findings may be limited because: (a) differences between groups in the AS task may be driven by other aspects of MS, rather than deficits in inhibitory control per se; and (b) is unclear that AS measures are selective for MS per se, or if the same differences would be present in other comorbid conditions (excluded from the tested population). Concerning (a), we performed a comprehensive characterization of the neurocognitive performance in the MS group and measured its association with the results of Stroop and the AS task. These experimental decisions were made to minimise this possible limitation imposed by multiples deficits caused by MS. Concerning (b), comorbid conditions that can interfere with AS performance were excluded in the sample tested.

## CONCLUSIONS

Our findings suggest that AS may be a selective and independent measure to investigate inhibitory control in patients with MS. More studies are necessary to confirm our results and to describe brain correlates associated with impaired performance in the AS task in people diagnosed with MS.

## ACKNOWLEDGEMENTS

We would like to thank the Clinical and Academic Centre (2CA-Hospital de Braga) for providing facilities for data collection. We also want to acknowledge: (i) Carla Sofia Ferreira for scheduling all the MS participants and most of the controls; (ii) one anonymous reviewer for his opinion about an early version of this manuscript; and (iii) Dora Marques for her help with data collection. Parts of this work have been presented at 3rd International Porto Congress of Multiple Sclerosis, February 27–28, 2015, Porto, Portugal and ARVO 2015 annual meeting, May 3–7, Denver, Colorado. In ARVO

2015 the number of participants was different because the neuropsychological tests were only performed in a sub-group of participants with MS.

### Funding

The Vision Rehabilitation Lab and Antonio Filipe Macedo receive or received funding from Shamir Optical Industry Lt, Portugal, from grants PTDC/DTP-EPI/0412/2012 (Prevalence and Costs of Visual Impairment in Portugal) and UID/FIS/04650/2013 (Framework of the Strategic Funding granted to Centre of Physics at Minho University). Multiple Sclerosis Association "Todos com a Esclerose Multiple" paid the salary of Marisa Borges Ferreira. The funders had no role in study design, data collection and analysis, decision to publish, or preparation of the manuscript.

### Grant Disclosures

The following grant information was disclosed by the authors:
Shamir Optical Industry Lt, Portugal, from grants PTDC/DTP-EPI/0412/2012 (Prevalence and Costs of Visual Impairment in Portugal) and UID/FIS/04650/2013 (Framework of the Strategic Funding granted to Centre of Physics at Minho University).
Multiple Sclerosis Association "Todos com a Esclerose Multiple" paid the salary of Marisa Borges Ferreira.

### Competing Interests

The authors declare that they have no competing interests.

### Author Contributions

- Marisa Borges Ferreira conceived and designed the experiments, performed the experiments, analysed the data, prepared figures and/or tables, authored or reviewed drafts of the paper, approved the final draft.
- Paulo Alexandre Pereira conceived and designed the experiments, authored or reviewed drafts of the paper, approved the final draft, funding.
- Marta Parreira conceived and designed the experiments, performed the experiments, analysed the data, authored or reviewed drafts of the paper, approved the final draft.
- Ines Sousa conceived and designed the experiments, analysed the data, authored or reviewed drafts of the paper, approved the final draft.
- José Figueiredo Figueiredo conceived and designed the experiments, authored or reviewed drafts of the paper, approved the final draft.
- João José Cerqueira conceived and designed the experiments, contributed reagents/materials/analysis tools, authored or reviewed drafts of the paper, approved the final draft, recruitment and selection.
- Antonio Filipe Macedo conceived and designed the experiments, performed the experiments, analyzed the data, prepared figures and/or tables, authored or reviewed drafts of the paper, approved the final draft.

## Human Ethics

The following information was supplied relating to ethical approvals (i.e., approving body and any reference numbers):

The ethics committee at University of Minho granted ethical approval to carry out the study, ref: SECVS 083-2013.

## Data Availability

The raw data and R code are provided in the Supplemental Files.

## Supplemental Information

Supplemental information for this article can be found online at http://dx.doi.org/10.7717/peerj.5737#supplemental-information.

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
