# Peer review of "Relationships between neuropsychological and antisaccade measures in multiple sclerosis patients"

_PeerJ, doi:10.7717/peerj.5737_

## Round 0.1 · original submission · Minor Revisions

Two reviewers have read your manuscript and both feel that it requires minor revisions. Please address each concern fully and provide a line-by-line response.

Reviewer 1 ·

Basic reporting

Line 292 “less prone to be affected by confounding cognitive, visual and socio-demographic aspects” needs citations, because I suspect that this may be incorrect, and that AS task performance might be confounded by cognitive, visual, and socio-demographic aspects. Certainly AS task performance is altered by cognitive deficits, as pointed out in line 310 of the present manuscript: slow cognitive responses are expected to cause poor performance in the AS task. Visual factors might affect antisaccade tasks: if a participant cannot see the fixation cross well, it may be more difficult to make fast and accurate saccades to the mirrored location. Socio-demographic aspects may relate to a participant’s ability to clearly understand the (somewhat complex) task instructions involved in looking quickly and precisely to the mirrored direction of a cued location.

Minor comments:
The language is detailed, professional, and generally clear. The phrasing of lines 113-114 and 182-184 should be improved.
Line 38: “i) to investigate if antisaccades performance in patients with multiple sclerosis” is not a complete thought. Investigate if they what?
Line 95: “Some studies that looked” should be “Some studies looked”
Line 114 “poor readers may have less interference in the Stroop test than good readers” needs a citation, and/or further justification.
Line 130 “was also recruited” should be “were also recruited”

Experimental design

I have few concerns about the study design.
Was the test battery administered to the MS group before or after the antisaccade task? This is not clear from the manuscript. The test battery should have been administered after (or on a separate occasion), as the antisaccade task results might be altered by fatigue: the MS group, fatigued by the test battery, would then have more errors or longer latencies on the antisaccade task, relative to the control group (who did not take the test battery and were therefore not fatigued). Differences could no longer be clearly attributed to MS, as they might be due to fatigue.
Those with “clinically visible oculomotor abnormalities (e.g. nystagmus or internuclear ophthalmoplegia)” were excluded. It would be best to provide a full list/table of the abnormalities that were cause for exclusion—and to provide the number of potential subjects that were excluded—as the antisaccade test may potentially be invalid for similar patients in the future. Just as the Stroop test cannot be used in colorblind patients, the antisaccade task cannot be used in patients who have known saccadic abnormalities for reasons unrelated to MS. Because the motivation behind the current study is that the Stroop test is invalid in some cases, the cases where this other test is also likely to be invalid should be made clear.
Line 152: “One or two blocks of 8 practice trials” How was the number of blocks of practice determined? Did the MS group receive more or less practice (on average) than the control group? If the practice was not equivalent between groups, this may have led to some differences between the groups.

Minor comments:
What version of Matlab was used?
What size (dva) were the crosses and the gaze-contingent box? These details should be included in the manuscript. I believe that the figure is not to scale, though this is not made explicit.
Were the crosses and the gaze-contingent box white or yellow? They appear yellow in the figure.

Validity of the findings

The findings appear sound, and conclusions are generally appropriate.
The authors conclude that the present findings show antisaccades to be “selective” for inhibitory control in MS, and it is not clear to me how this has been demonstrated. The present results demonstrate that the antisaccade task outcomes are altered by MS, even in early stages of disease severity. But other aspects of MS may drive these differences, rather than deficits in inhibitory control per se. Further, it is not clear that these measures are selective for MS per se, or if the same differences would be present in other comorbid conditions (excluded from the tested population).
It is unclear why statistical comparisons between the MS and control group for prosaccade latencies and corrective saccade latencies are not presented. Similarly, the AS peak-velocity section does not discuss any comparisons between the MS group and controls, despite the stated aims of the study. Corrective saccades should reach a higher peak velocity than antisaccades, as they need to travel a larger distance (and saccades follow the main sequence). This unsurprising finding was confirmed, but no MS vs control comparison is presented for peak velocity, including in the table of correlations with the Stroop test.
The raw data are provided, can be opened, but are not well described. What do the different “sac_type” values correspond to? Which eccentricity is “trial_ecc” 1 and which eccentricity is “trial_ecc” 2? Is “trial_direction” 1 to the left or the right? I thank you for providing the raw data, however your supplemental files need more descriptive metadata identifiers to be useful to future readers.

Reviewer 2 ·

Basic reporting

No comment.

Experimental design

No comment.

Validity of the findings

No comment.

Additional comments

The authors have performed detailed analyses of antisaccade (AS) task performance, a widely studied measure of oculomotor movement, in patients with multiple sclerosis (MS) and age- and gender-matched controls. A key premise of the current study is that the AS task may distinguish MS cases to a comparable degree as the standard Stroop interference test, while being more robust to nonspecific effects of neurocognitive functioning. Key analyses include tests of types of saccade (prosaccade, corrective, and antisaccade) in cases and controls, and correlations of AS latency with neurocognitive performance in cases. Overall, the paper is clearly written, the hypotheses are clearly stated, and adequate experimental detail is provided. Our main concern is the rather limited discussion of the potential implications of this work, namely with respect to the relevance of inhibitory control in the onset and progression of MS, and the potential of these findings to inform clinical practice. Some other comments are detailed below.

# MAIN SUGGESTIONS
- some commentary on the practicality of the AS task as an alternative to the Stroop is warranted (e.g. in terms of adminstritation of the test, required equipment, etc), as especially in terms of clinical utility.
- are "normative" Stroop results available? Couldn't the magnitude of these deficits (compared to healthy controls) be summarized here, as you show for the neuropsychological tests?
- at line 176-178, the authors state that "all neuropsychological tests have normative data for the Portuguese population". This is taken as implying that the these tests have been translated into the Portuguese language and administered to native Portuguese speakers, with results consistent to those observed for native English speakers. This is important to note, but really speaks only to the validity of these tests when translated from English. This doesn't speak to the extent of inter-individual variability in any given population, nor are these results a substitute for measuring neurocognitive performance in controls.
- did you correct for age and gender in your analyses? It is understood that controls were age and gender matched, but this is not equivalent to accounting for possible performance differences among people aged 18-45, etc.
- Table 4: the use of "ns" instead of displaying non-significant p-values (and correlations) is not widely preferred (there is a difference between 0.056 and 0.95), inconsistent with Table 2 (you don't do this here), and frankly a bit lazy. You might also want to spell out this abbreviation under the first table to use these.
- lines 113-115: please provide a citation for limitations of Stroop test with respect to inhibitory control, i.e. difference in performance among "good" and "poor" readers.
- Figure 1B: what are the thinner red lines here? These are not explained and as such only detract from the readability of the figure.


# MINOR ISSUES
- line 148: please spell out the first use of "SDK" (software development kit?)
- the use of commas throughout the manuscript is somewhat erratic
- line 174: "fonetic" should be "phonetic" (Portuguese vs English spellings...)
- lines 68-69: typo "this his task"
- line 183: should be phrased as "can lead to different and somewhat unexpected results"
- line 212: incomplete sentence; should be "Bonferroni correction *was applied* for multiple comparisons"
- line 254: should read "A model similar to *that* reported for latency"
- Table 2: "Trail making Man" should be "Trail making Men" ("Trail making (Men)" is preferred)

---

## Round 0.2 · accepted · Accept

Thanks for updating your manuscript, it looks great and I am recommending to accept for publication.

#